# Observations of marine animal interactions with a small tidal turbine

Emma Cotter[1,2]*, Christopher Bassett[3], Paul Murphy[4], Mitchell Scott[4], Alexa Runyan[4], Jood M. Almokharrak[3], Lucy G. Kao[3], Lillian M. Ovall[4], Suni A. McMillen[3]

**1** Coastal Sciences Division, Pacific Northwest National Laboratory, Sequim, Washington, United States of America, **2** Department of Mechanical Engineering, University of Washington, Seattle, Washington, United States of America, **3** Applied Physics Laboratory, University of Washington, Seattle, Washington, United States of America, **4** MarineSitu, Seattle, Washington, United States of America

* emma.cotter@pnnl.gov

## Abstract

The risk of collisions between animals and operating tidal turbines remains a concern in the scientific and regulatory communities. A sensor package including optical cameras was deployed to monitor animal interactions with a small-scale (1 m$^2$) cross-flow tidal turbine. The turbine was deployed in Washington State, USA for 141 days at a site with peak flow speeds of 2.5 m/s. We analyze optical camera imagery spanning 109 days of turbine operation. The analyzed images contain 1044 observations of fish, fish schools, seabirds, or seals in the vicinity of the turbine. No instances of collision with seabirds or seals were observed. Seabirds were only observed during daylight hours and while the turbine was stationary. Both seals and fish were observed during both day and night and while the turbine was stationary and rotating. Four fish were observed colliding with the moving turbine and in all but one case the animals swam away following the collision. Over the same period of time, over fifty times more fish (224 individual fish and 5 fish schools) were observed passing the moving turbine without collision. Fish encounters were likely under counted due to the difficulty in discerning small fish from plant matter in the water column. These observations represent the first optical camera imagery showing fish, bird, and marine mammal interactions with a tidal turbine in North America. In addition to quantitative and qualitative discussion of the implications of our observations for collision risk, we discuss lessons learned on sampling schemes and deployment of machine learning for detection of animals to inform future data collection strategies in future monitoring campaigns.

## Introduction

In recent years, several projects have successfully demonstrated the feasibility of electrical power generation from tidal currents at grid scale [1]. At smaller scales, there is interest in leveraging analogous technologies for generating power at sea for

**Data availability statement:** Video data are available on the Marine and Hydrokinetic Data Repository here: https://mhkdr.openei.org/submissions/599. Metadata are included as Supporting information files.

**Funding:** The development and deployment of the Turbine Lander was sponsored by the Naval Facilities Engineering and Expeditionary Warfare Center (NAVFAC) under Naval Sea Systems Command contract N00024-20-F8708. Environmental monitoring data management was funded by the U.S. Department of Energy Water Power Technologies Office (WPTO) TEAMER program, and analysis of the collected data was funded by WPTO (award EE0007827). Development of the machine learning and data management tools used by MarineSitu for this work was supported by WPTO through the Small Business Innovation and Research project "Modular Instrumentation and Automated Data Processing for Marine Energy Monitoring" (Award DE-SC0021845). The funder provided support in the form of salaries for all authors, but did not have any additional role in the study design, data collection and analysis, decision to publish, or preparation of the manuscript. The specific roles of these authors are articulated in the 'author contributions' section.

**Competing interests:** Authors P.M., A.R., and L.O. are employed by MarineSitu or were employed by MarineSitu at the time of the study. This does not alter our adherence to PLOS One policies on sharing data and materials.

applications including scientific sampling, autonomous vehicle recharge, and aquaculture [2,3]. However, uncertainty surrounding the potential effects on marine animals remains a barrier to development of both grid- and small-scale tidal energy [4–7]. Potential effects include electromagnetic fields, underwater sound, physical effects on the surrounding environment (e.g., scour), and animal injury or mortality associated with collisions between animals and operating tidal turbines [6,7]. Notably, the limited volume of data to inform assessments of the risk of collision to fish, marine mammals, and diving seabirds has hindered tidal energy consenting processes globally.

Garavelli et al. [6] defines several terms that describe how animals might behave in the vicinity of a tidal turbine that will be used throughout this paper. An animal is deemed to have *avoided* a turbine if it responds to the presence of a turbine and moves away from it at a distance greater than 5 times the turbine diameter. Conversely, an animal *encounters* a turbine if it comes within a range of 5 turbine diameters. If an animal encounters a turbine, it might *evade* the turbine (i.e., change its behavior to avoid contact with the turbine) or experience *collision* (i.e., come in contact with a moving component of the turbine). While there have been no observations of collision between fish, marine mammals, or diving seabirds and tidal turbines reported in the literature to date, previous studies offer some insight into animal behavior around turbines.

In several studies, fish have been observed avoiding tidal turbines while they are operating [8–11], and evasion behavior by those fish that do encounter a tidal turbine has also been observed [9,11–13]. Based on a mobile echosounder survey, Grippo et al. [8] observed a decrease in fish abundance within 140 m of a tidal turbine deployed in Cobscook Bay, Maine (USA) while it was operating (avoidance), and this decrease was not observed while the turbine was stationary. Both Bevelheimer et al. [13] and Viehman and Zydlewski [12] used acoustic cameras to monitor fish activity in the vicinity of tidal turbines, and both observed decreased fish presence while the turbine was operating (i.e., avoidance) as well as instances of evasion behavior. In a fish release study conducted in a river in Sweden, Bender et al. [10] used an acoustic camera to observe that brown trout (*Salmo trutta*) rarely approached a turbine, regardless of operational state, and Atlantic salmon (*Salmo salar*) maintained a greater distance from the turbine while it was rotating. Similarly, Hammar et al. [11] used optical cameras to monitor a small tidal turbine, and observed fewer fish in the area when the turbine was operating (avoidance) as well as evasion behavior by several species that did encounter the turbine. Finally, in 4,000 hours of data recorded by motion-activated optical cameras mounted on a tidal turbine in Bluemull Sound (UK), 28 hours of which were acquired while the turbine was operating, Smith [9] observed fewer saithe (*Pollachius virens*) around the turbine during strong currents, and reported five instances of fish evading the moving turbine blades. No instances of fish passing through the rotor's swept area were reported.

Due to the relatively low abundance of marine mammals compared to fish, there have been fewer observations of marine mammal behavior around tidal turbines. Harbor seals (*Phoca vitulina*) were observed in the vicinity of the tidal turbine monitored in Smith [9], but they were rare (only 10 instances observed) and were never

observed when the turbine was operating. Using an array of hydrophones mounted to the base of a turbine deployed in Pentland Firth, UK, Gillespie et al. [14] found that harbor porpoises (*Phocoena phocoena*) that encountered a turbine effectively evaded the turbine blades, and only a single porpoise passed through the swept area (while the turbine was stationary) during 451 days of monitoring. Further analysis of the same data in Palmer et al. [15] indicates that harbor porpoises also avoided the turbine area while it was operating. Avoidance of turbines has also been demonstrated for harbor seals. At the same site as Gillespie et al. [14] and Palmer et al. [15], a multibeam sonar was used to study seal presence within tens of meters from a turbine, and found that fewer seals were present at flow speeds above the 1.2 m/s threshold for turbine operation. This number decreased even further when the turbine was operational [16]. Onoufriou et al. [17] observed a decrease in harbor seal abundance within 2 km of an array of tidal turbines while they were operating, and Hastie et al. [18] demonstrated an 11-41% reduction in seal presence near an acoustic projector simulating the sound of an operational tidal turbine.

Collision with turbines also presents a risk to diving seabirds, but there have been limited observations of seabirds around tidal turbines to date. Smith [9] observed European shag (*Phalacrocorax aristotelis*) and black guillemots (*Cepphus grylle*) in the vicinity of the monitored turbine. As with seals, these observations were rare (15 instances observed) and seabirds were only observed during periods when the turbine was not operating. Others have studied seabird habitat use in areas suitable for tidal energy development [19–21], but, to our knowledge, the only direct observations of seabird interactions with an operational tidal turbine to date are those described in Smith [9].

While existing studies indicate that many animals avoid the area surrounding operational tidal turbines, and some animals that do encounter tidal turbines are capable of evasion, the risk of collision remains a key concern. Further, most studies to date have investigated animal avoidance at larger scales, but there have been few reports of fine-scale evasion behavior in the immediate vicinity of a tidal turbine. In this work, we present observations made over several months of harbor seal, seabird, and fish encounters with a small tidal turbine designed to provide power for at-sea operations. Throughout the turbine deployment, sampling strategies were adjusted based on observations in the collected data and lessons learned. While this limits our ability to quantitatively evaluate animal encounter rates or long-term trends in animal presence, analysis of collected data provides insight into how different species behave in the presence of an operational tidal turbine. We also discuss lessons learned for effective data collection, automated data processing, and analysis to inform future monitoring campaigns.

## Methods

### Turbine lander

The Turbine Lander (Fig 1) is a marine energy converter system that includes a vertical axis, cross-flow turbine on a gravity foundation. The four-bladed rotor is 1.19 m tall and 0.85 m in diameter with blade chord lengths of 10.2 cm. The individual blades have a wet weight of approximately 1.5 kg and the turbine's moment of inertia is estimated to be approximately 2 kg m$^2$. When deployed, the bottom and top of the rotor are approximately 1.5 and 2.7 m above the seabed, respectively. While small, the system is considered full-scale as the target application is to provide modest amounts of power at sea where no cabled infrastructure exists. Environmental monitoring equipment (the Adaptable Monitoring Package) was distributed on several areas on the foundation.

During the deployment analyzed in this study, the turbine was primarily operated in speed control such that it maintained a constant tip-speed ratio near 2 (i.e., the tangential velocities of the rotor blades were approximately twice as fast as the inflow current speed). The minimum flow speed for turbine operation ("cut-in speed") was adjusted between 0.9 m/s and 1.0 m/s at different times throughout the deployment. Below the cut-in speed, the turbine rotor was programmed to be stationary. More information about the Turbine Lander's design and operation is can be found in Bassett et al. [22].

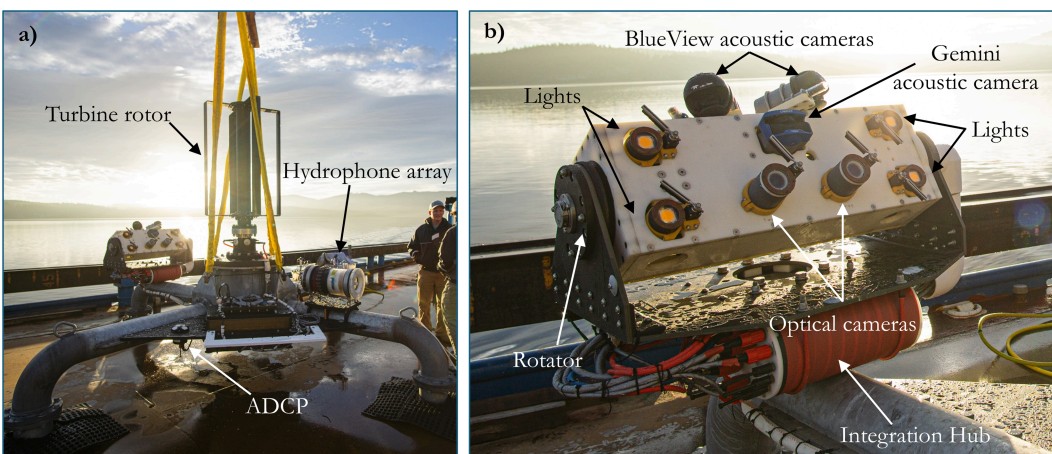

**Fig 1**. **Turbine and monitoring equipment**. (a) The Turbine Lander prepared for deployment. (b) The Adaptable Monitoring Package instrument head with sensors annotated. Photo credit: Abigale Snortland.

## Test site

The Turbine Lander was deployed on 18 October 2023 in the ~250 m wide tidal channel at the entrance to Sequim Bay, Washington, USA at 46° 4.761'N, 123° 2.589'W, adjacent to Pacific Northwest National Laboratory's (PNNL) Marine and Coastal Research Laboratory (Fig 2). Bathymetric data in Fig 2 were collected by C & C Technologies Survey Services and provided by Pacific Northwest National Laboratory. The deployment location is approximately 8 m deep at mean lower low water (MLLW) and characterized by mixed semi-diurnal tides that result in peak tidal velocities of approximately 2.5 m/s. The system was recovered on 7 March 2024, but the turbine was not operated after 11 February due to the

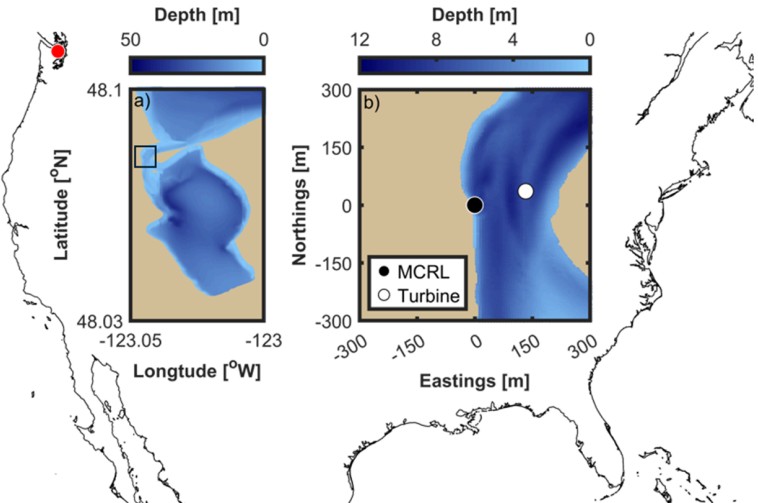

**Fig 2**. **Study site**. The background outlines central North America with the test site at the inlet to Sequim Bay, Washington (USA) highlighted by the red dot (upper right corner). (a) The bathymetry (MLLW) of Sequim Bay. The box at the constriction at the north end of the bay highlights the inlet, which is shown in (b). (b) The bathymetry of the inlet to Sequim Bay showing the locations of the PNNL Marine and Coastal Research Laboratory (MCRL) dock and the deployment location of the Turbine Lander and the Adaptable Monitoring Package (AMP).

failure of two blades (see [22] for details). The system was cabled to shore, where shore-side computers handled environmental data acquisition and turbine control.

Sequim Bay is a habitat for fish, marine mammals, and seabirds [23]. Harbor seals are frequently observed in the area. Since 2021, more than 15 species of diving seabirds have been reported in the vicinity of PNNL-Sequim, including several species of waterfowl, three species of cormorant, and two species of auk [23]. Several forage fish species including Pacific herring (*Clupea pallasi*), surf smelt (*Hypomesus pretiosus*), and Pacific sand lance (*Ammodytes personatus*) are known to spawn in Sequim Bay, and are the most abundant forage fish species in the region [24,25]. Multiple species of salmonids and rockfish are also found in the area, with several species listed as endangered or threatened [23]. Among other species commonly observed in the area are sculpin, snailfish, flatfish, and perch.

## Adaptable monitoring package

An integrated sensor system, the Adaptable Monitoring Package (AMP), which is described in detail in Polagye et al. [26] and available commercially through MarineSitu, was installed on one leg of the Turbine Lander platform to monitor animal activity in the vicinity of the turbine (Fig 1). The AMP included an optical camera system and three acoustic cameras. The optical camera system consisted of two Allied Vision Technologies Manta G-507b board level cameras in custom housings with KOWA lenses (LM5JC1M, 82° field of view). Four custom-built strobe lights that contained Cree CXB 3590 white LEDs provided artificial illumination for the optical cameras. The acoustic cameras on the AMP included two Teledyne BlueViews (M900-2250) and a Tritech Gemini (720is). The BlueView acoustic cameras were both configured with a 10 m maximum range and the Gemini was configured with a 15 m maximum range. The BlueView acoustic cameras were operated in their higher frequency mode (2250 kHz). To facilitate imaging of animals both up and downstream of the turbine during both flood and ebb tides, the BlueViews were oriented perpendicular to each other such that their beams overlapped by approximately 45° at the rotor. The Gemini was oriented such that the rotor was centered in its field of view. The AMP software triggered acquisition of the acoustic images with short time delays (e.g., < 50 ms) to avoid crosstalk while minimizing elapsed time between images.

Optical and active acoustic sensors were mounted on a tilt motor such that the field of view of the sensors could pan up and down in the water column. Throughout the deployment, the orientation was occasionally shifted up or down during system health checks, but the full turbine rotor was maintained within the field of view except for these short manual interventions (no more than a few minutes). In addition to the sensors on the AMP, a small passive acoustic array consisting of four hydrophones (custom data acquisition, HTI 90-HF hydrophones) was installed on one leg of the foundation and an acoustic Doppler current profiler (ADCP; Nortek Signature1000) was installed on a mounting plate between two legs of the foundation to measure flow speeds.

In this study, we focus on data from the optical cameras. Data from the acoustic cameras are referred to for additional context, and data from the ADCP are used to evaluate trends in animal presence with tidal currents. A detailed description of the acoustic camera data can be found in Bassett and Cotter [27].

## Optical camera datasets

Two optical camera data acquisition strategies were used during the deployment: 1) scheduled data acquisition, and 2) data acquisition when animals were predicted to be present by real-time detection models operating on either the optical camera or acoustic camera data. Data acquisition methods varied over the course of the deployment as collected data were reviewed and informed more optimal approaches. Lessons learned are discussed in detail in the Discussion section.

Scheduled data were either acquired on a user-specified duty cycle (e.g., 5 s every 1 min) at a high frame rate (20-24 Hz) or continuously at 1 Hz. The strobe lights were activated during duty cycle data acquisition, but not during

continuous data acquisition. Continuous optical camera data collection was never implemented at night when artificial illumination would have been required because of concerns that continuous operation of the strobe lights would influence animal behavior.

Detection models operated in real time on each image by the cameras. When the models predicted that an animal was present (i.e., detected an animal in the image), data from 5 s before and 5 s after the frame detected by the model were archived with a frame rate of at least 20 Hz. Additionally, strobe lights were activated for 2.5 s following each model detection. We call each archived window of data a "detected event".

Both optical camera and acoustic camera models were developed using MarineSitu's commercial license of the Ultralytics YOLO model [28] and developed using MarineSitu deployment software. Models were iteratively trained and redeployed as more images were acquired and annotated throughout the deployment. The optical camera detection model was trained to detect seals, diving seabirds, and fish, however it only reliably detected seals and diving seabirds, possibly due to the fact that large fish were infrequently observed and, in many cases, small fish appeared similar to drifting debris, especially at farther ranges from the cameras. Because illumination was not constant, optical camera models were only able to detect animals during daylight hours or when the strobe lights were illuminated on a duty cycle. When detection occurred at night during duty cycle acquisition, artificial illumination was activated after each frame in which the animal was detected (i.e., the detected event might be longer than the scheduled duty cycle window).

Acoustic camera detection models (only deployed after 1 February) were trained to detect seals and diving birds, but did not perform well. During some periods, the model was tuned too permissively such that nearly all detections were false positives resulting from detritus in the water column or reflections from the moving turbine (e.g., 34,287 detections on the night of 2 February). During other periods, the acoustic camera model was tuned too conservatively such that it did not capture most animals of interest (e.g. 64 detections on the night of 10 February). Despite this poor performance, optical camera data collected as a result of either correct or incorrect detections from the acoustic camera model are included in our analysis because they contained many interesting observations of animal behavior around the moving turbine at night, when our sampling was otherwise limited.

We analyzed all optical camera images collected during periods of continuous recording and all detected events from 1 November 2023 to 17 February 2024, five days after the turbine was shut down. Because data collection methods were varied, these data are broken into three distinct subsets for analysis, which are indicated in Fig 3 and are described in the subsequent paragraphs. The first two subsets are comprised of scheduled data collection (duty cycle and/or continuous acquisition), and the third subset is comprised of detected events. Table 1 provides an overview of the duration of data recorded in each data subset, including the total number of hours in the sampling period, the total number of hours in the sampling period when the turbine was operational, the total number of hours with optical camera data recorded at a frame rate of at least 1 Hz, and the total number of hours with optical camera data recorded at a frame rate of at least 1 Hz while the turbine was operational.

Data subset 1 consists of scheduled data collection from 1-8 November. Data were acquired continuously at a frame rate of 1 Hz during the day, and for 3 s every 1 min with artificial illumination at night. No data were collected on 17 November between 13:04 and 18:00 (local time) when the turbine and AMP were shut down for maintenance.

Data subset 2 consists of scheduled data collection between 25 January and 17 February. Data were collected at 1 Hz during daylight hours, and no scheduled data collection was employed at night. The turbine was no longer operating during the last 5.5 days of this review period (after 11 February at 4:30 local time), and no data were collected on 6 February from 9:21 to 14:18 (local time) during system maintenance. We note that during collection of both data subset 1 and data subset 2, occasional network interruptions resulted in lost frames (i.e., frame rate < 1 Hz).

Data subset 3 is comprised of all detected events from the automatic detection models from 1 November to 17 February. Detection models were disabled from 8-17 November, so no events captured in data subset 1 are contained in data subset 3. Conversely, detection models were operating in concert with continuous data acquisition in data subset 2, so

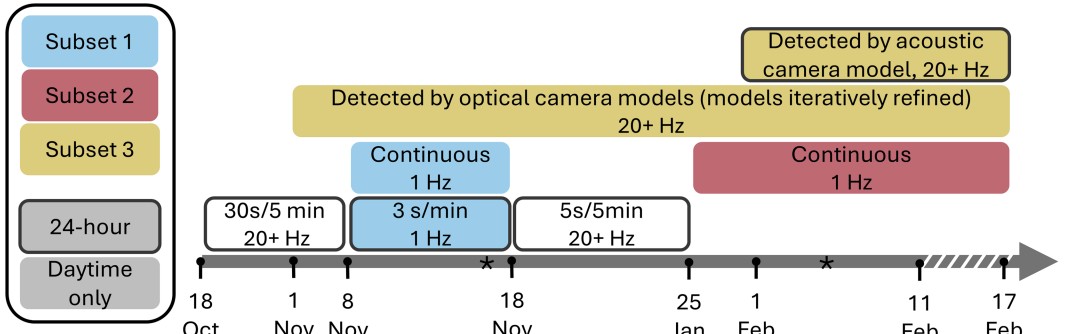

**Fig 3. Timeline summarizing data collection.** The three data subsets that were analyzed are indicated. Periods with a dark border include nighttime data, while periods without a dark border were only collected during daylight hours. Periods of data in white were not analyzed, and asterisks (*) indicate periods where the AMP was offline for up to 5 hours for system maintenance. The dashed portion of the timeline after 11 February indicates the period when the turbine was deployed, but not operating. Note that the timeline is not to scale.

**Table 1. Data summary.** Summary of the data contained in each subset, including the total number of hours contained in each sampling window, the total number of hours in the sampling window when the turbine was operating, the total number of hours of recorded data at at least 1 Hz, and the total number of hours of data recorded at a frame rate of at least 1 Hz while the turbine was rotating.

|  | Total Hours | Total Hours with Turbine Operating | Hours Recorded | Hours Recorded with Turbine Rotating |
|---|---|---|---|---|
| Subset 1 | 240.0 | 88.8 | 81.5 | 8.0 |
| Subset 2 | 552.0 | 126.3 | 208.9 | 28.6 |
| Subset 3 | 2616.0 | 776.8 | 61.1 | 21.4 |

events contained in data subset 2 that were automatically detected are also contained in data subset 3. However, only data subset 3 contains high frame rate recordings of these events.

Even though the automatic detection models did not effectively detect fish, many interesting observations of fish are included in data subset 3. These events were largely captured due to false positive detections by the detection models (e.g., the model detected a piece of kelp entrained on the turbine). We include these observations in this paper because they represent some of the first observations of fine-scale fish behavior around a moving tidal turbine. However, while our observations of fish offer insights into fish behavior around a moving tidal turbine, they cannot be used to quantitatively assess the number of fish that encountered the turbine or the probability of collision.

## Data review and analysis

Every optical camera image included in data subsets 1, 2, and 3 was initially reviewed by at least one member of team of five human reviewers (authors C.B., J.A., L.K., L.O., and S.M.). We grouped observations into four classes for analysis: individual fish, fish schools, seabirds, and seals (harbor seals were the only observed species of marine mammal). Whenever one of the four classes was observed, the timestamps of the first and last images in which the animal or fish school was visible were logged as an event. During initial review, reviewers included any events containing ambiguous targets that could potentially be animals in the log. In some cases, an animal was not classifiable in an individual frame (e.g., edge of a flipper is the only part of a seal in view), but was classified based on observations in other images. Fish schools were logged as a single event when individual fish within the school were too numerous to reasonably count. With the exception of fish schools, two separate events were logged when two animals of interest were simultaneously detected, and if the same animal left the frame and then reentered, one event was logged. This required some subjective interpretation as seabirds and seals regularly left the frame and then reentered shortly thereafter. For seals, a threshold of 60 s was used to determine if a new event should be logged. The same threshold was used for seabirds unless it was

observed swimming to the surface and another dive was observed within the 60 s threshold (this would be logged as two events). Reviewers indicated whether each event was recorded during dark or light conditions, which was distinguished by whether or not artificial illumination was deemed necessary to observe the animal. This also required some subjectivity for events captured during dawn or twilight when light conditions were rapidly changing and artificial illumination was used but may not have been required to identify the animal. Initial annotation of events was conducted by five separate reviewers.

After initial annotation, each event was analyzed in detail by a single reviewer (C.B.). Events appearing in more than one data subset were consolidated such that they are represented in each subset in which they appeared, but the aggregated event counts presented in the Results section reflect only unique events. Taxonomic classification was performed to the lowest level possible for each animal, and qualitative observations of animal behavior were logged. These annotations are considered the final dataset that are discussed in the Results section and included in the Supplemental Data. Observations of fine-scale animal behavior, like evasion, were only possible for the events in data subset 3 due to the higher frame rate (20+ Hz). A more detailed description of this review process can be found in S4 Detailed Description of Human Review Process.

Lastly, concurrent acoustic camera data were reviewed for all events where animals were observed evading or colliding with the turbine. While the emphasis of this study is on the optical camera imagery, acoustic camera data were reviewed to determine if they provided additional context for these animal behaviors of interest.

After analysis of all annotated events, we evaluated trends in animal presence and behavior. Because fish were not automatically detected, our analysis of fish behavior is limited to qualitative observations. For seals and seabirds, which were detected by the real-time detection model, we analyze trends in animal presence with respect to flow speed, turbine operating state, and tidal elevation. Tidal elevation was measured by a tide gauge on the nearby research pier, and are referenced to the North American vertical datum of 1988 [29]. Flow speed was measured by the ADCP on the Turbine Lander, and was calculated as the 2-minute moving average at an elevation of 3.1 m above the seabed (approximately 40 cm above the top of the rotor). For each class, we determined the total number of events ($n$), the number observed when there was ambient light ($n_{light}$); the number observed when it was dark and artificial illumination was required ($n_{dark}$); the number observed when the turbine was rotating ($n_{turb}$); and the number observed when flow speeds exceeded the lowest turbine cut-in speed of 0.9 m/s ($n_{cutin}$). We note that $n_{cutin}$ and $n_{turb}$ differ because of tidal cycles when the turbine was not operational at the end of the deployment and one instance when the turbine was rotating during slack tide due to a communication failure with the ADCP.

Finally, we compared the trends in seal and bird observations in data subset 3 to those identified through human review of the continuously acquired data in subset 2 to assess model performance. Because the detection model was iteratively updated with newly acquired training data throughout the deployment, we cannot rigorously assess model performance or use the triggered events to quantitatively assess animal abundance or encounter rates. However, comparison with continuously acquired data offers some insight into whether the model reliably detected seals and seabirds.

## Results

Review of the images in the three data subsets identified 1044 distinct events containing observations of fish, seals, or seabirds. While our analysis focuses on fish, seals, and seabirds due to regulatory interest, we note that kelp crabs (*Pugettia producta*), jellyfish, shrimp, euphausiids (krill), and sea slugs (order Nudibranchia) were also observed in the imagery. An overview of observed taxa, performed at the lowest level possible given the quality of the images, is included in S1 Species Table. The achieved level of taxonomic identification varied depending on ambient light, range to the animal, water clarity, and animal size. Species-level identification was straightforward for harbor seals, sometimes difficult for similar bird species, and typically not possible for small fishes. Water clarity varied throughout the deployment, which affected the ability to detect animals that were farther from the cameras than the turbine. In some conditions, the water

surface was visible (4+ m from the cameras), while at other times, the top of the turbine was not well defined despite being less than 2 m from the cameras.

An overview of the total number of events corresponding to each class in each data subset is included in Table 2. Fig 4 shows the distribution of events containing seals and seabirds in each data subset with respect to time of day, flow speed, and tidal elevation with the distribution of environmental conditions during all times with reviewed images in each data subset indicated for comparison. A representative time series of events from 2-8 February 2024 is shown in Fig 5. Among the patterns present in this time series are that bird detections were most common at high tides during daylight hours and that fish and seals were observed under a broad range of conditions. These trends are discussed in the subsequent sections and a full list of all annotated events, from all three data subsets, can be found in S2 Event Data. Below, we provide qualitative descriptions of the observations of each animal class. Videos of all fish collision and evasion events and representative bird and seal events are available online [30]. For a list of videos corresponding to each figure containing optical camera data, see S3 Videos.

### Individual fish

The annotated events contained 524 unique instances of individual fish (342 had high frame rate data). While discrimination between species of fish was difficult, no listed threatened or endangered fish species known to occur in the area (e.g., rockfish, salmonids, or sturgeon) were observed and identified. We note, however, that distinguishing Pacific eulachon (*Thaleichthys pacificus*), whose Southern Distinct Population Segment is listed as threatened, from other forage fishes is difficult. Thus, we cannot unambiguously state that they were not observed.

Reviewers identified 229 events containing fish encountering the turbine while it was rotating (Table 2). We note that identification of individual small fish in the imagery was challenging, particularly when fish were moving passively with the flow during periods with high volumes of plant matter in the water column, and only events where a confident classification could be made are included in this count. Fig 6 shows representative examples of fish detected while the turbine was operating. During periods of sufficiently strong flow for the turbine to be rotating, nearly all observed fish were classified as unidentified forage fishes or perch-like species, which are relatively small (<25 cm). Only two larger fish were observed while the turbine was rotating, and both were swimming high in the water column above the rotor and could not be identified (Fig 6b, 6c). While the turbine was rotating, fish were most commonly observed moving passively with the

**Table 2**. **Summary of all individual fish, fish schools, birds, and seals detected in data subsets 1, 2, and 3. Note that some events in data subset 2 are also contained in data subset 3.**

| | Subset 1 (81.5 hrs recorded) | | | | |
|---|---|---|---|---|---|
| | $n$ | $n_{light}$ | $n_{dark}$ | $n_{turb}$ | $n_{cutin}$ |
| Fish | 129 | 29 | 100 | 77 | 76 |
| Fish Sch. | 11 | 8 | 3 | 2 | 2 |
| Bird | 12 | 12 | 0 | 0 | 0 |
| Seal | 7 | 0 | 7 | 1 | 0 |
| | Subset 2 (208.9 hrs recorded) | | | | |
| | $n$ | $n_{light}$ | $n_{dark}$ | $n_{turb}$ | $n_{cutin}$ |
| Fish | 109 | 109 | – | 2 | 2 |
| Fish Sch. | 0 | 0 | – | 0 | 0 |
| Bird | 195 | 195 | – | 0 | 2 |
| Seal | 12 | 12 | – | 0 | 0 |
| | Subset 3 (61.1 hrs recorded) | | | | |
| | $n$ | $n_{light}$ | $n_{dark}$ | $n_{turb}$ | $n_{cutin}$ |
| Fish | 342 | 65 | 277 | 162 | 175 |
| Fish Sch. | 8 | 1 | 7 | 3 | 5 |
| Bird | 294 | 294 | 0 | 0 | 1 |
| Seal | 84 | 36 | 48 | 8 | 7 |

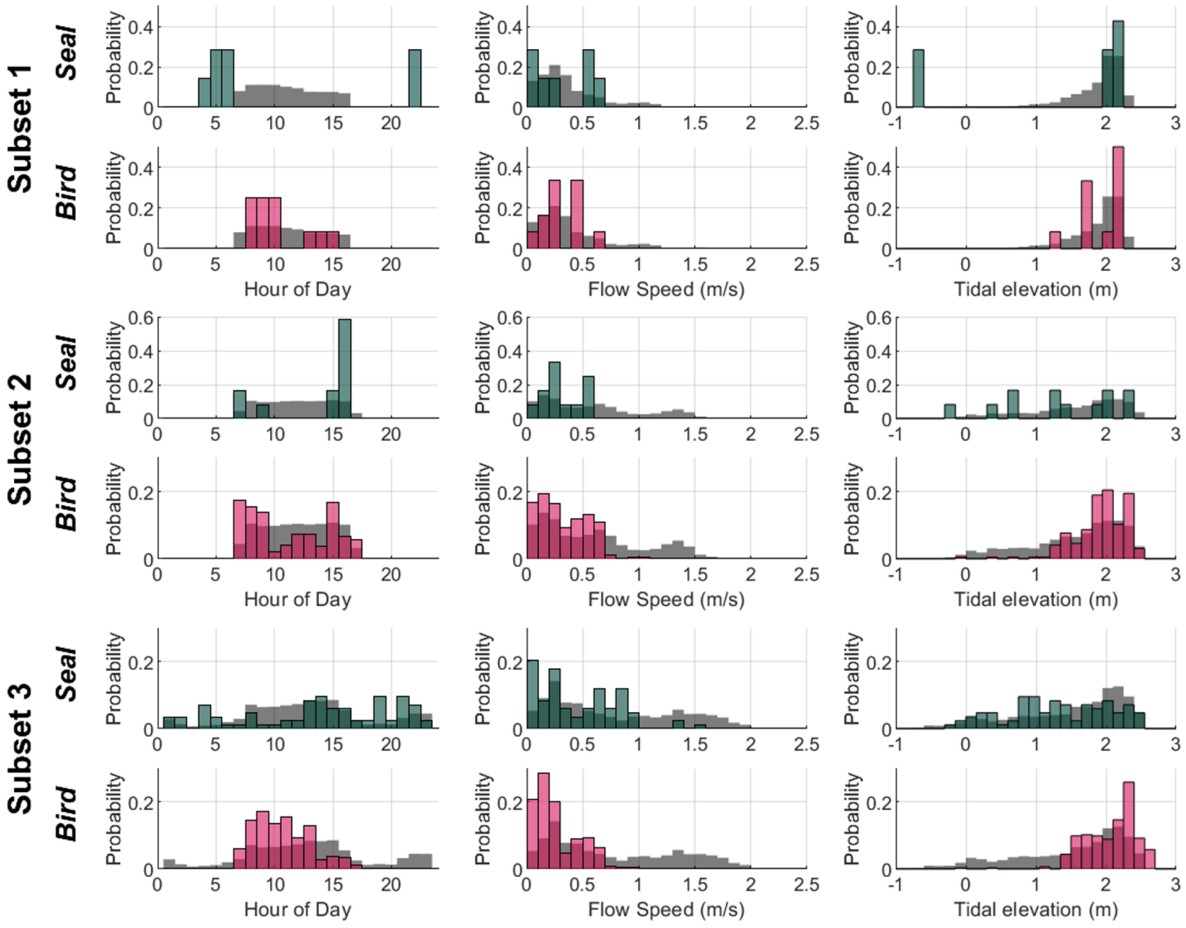

**Fig 4. Distribution of seal and bird events.** Distribution of events containing seals and seabirds in each data subset with respect to hour of day (local time), flow velocity, and tidal elevation (colored bars). Gray bars in the background of each histogram indicate the distribution of each environmental covariate throughout all images that were recorded and reviewed. Note that the y-axis scale differs between classes and data subsets.

flow (e.g., Fig 6a, 6d), although examples of fish swimming against the current or roughly perpendicular to it were also observed. We observed four instances of fish colliding with the moving turbine blades (1.7% of total identified fish encounters with the moving turbine). In 50 events (21.7%), fish exhibited behavior consistent with evasion (i.e., when on course to collide with the turbine or enter the area swept by the moving turbine blades, they made an apparent change in trajectory). The remaining 175 fish that encountered the turbine while it was rotating passed without collision, but were not categorized as evasion because they did not exhibit a discernible change in trajectory within the field of view of the cameras. These counts exclude events classified as fish schools, in which additional examples of evasion, but not collision, occurred.

All four observed instances of collision with the moving turbine involved relatively small fish (less than approximately 20 cm long based on their size relative to the blade's chord length). In three of the four events, the fish were oriented normally in the water column and swam away after collision (i.e., no mortality). An example fish collision is shown in Fig 7. The flow speeds during these collision events were 1.0, 0.9, and 2.0 m/s (blade speeds of approximately 2, 1.8, and 4 m/s), respectively, and fish were observed both moving with and against the currents before collision. In the fourth event, the fish did not appear to be moving after the collision and sank towards the seafloor out of view (i.e., potential mortality).

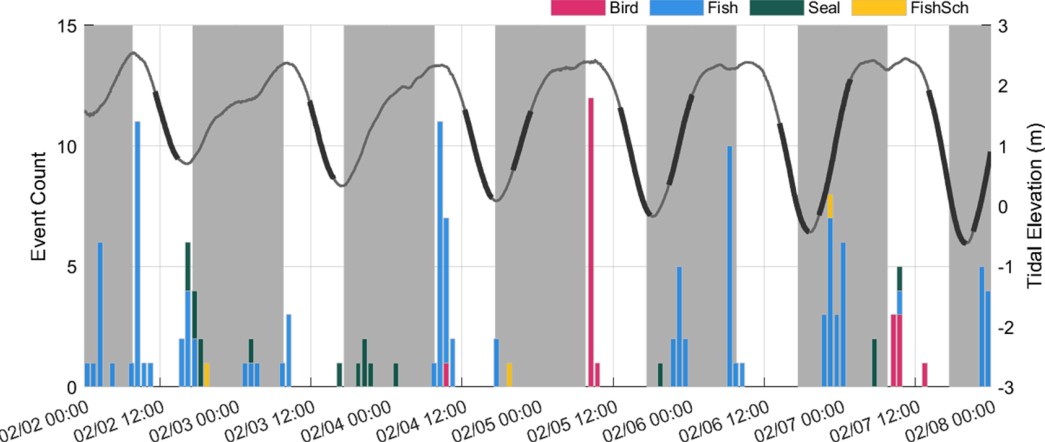

**Fig 5. Representative timeseries of events.** Number of events associated with each class per hour for a representative week of data. Events from both data subset 2 (continuous acquisition) and data subset 3 (detected events) are shown. Hours between sunset and sunrise are indicated in gray, and the tidal elevation is shown for reference. The darker regions of the tidal elevation time series indicate periods when the turbine was rotating.

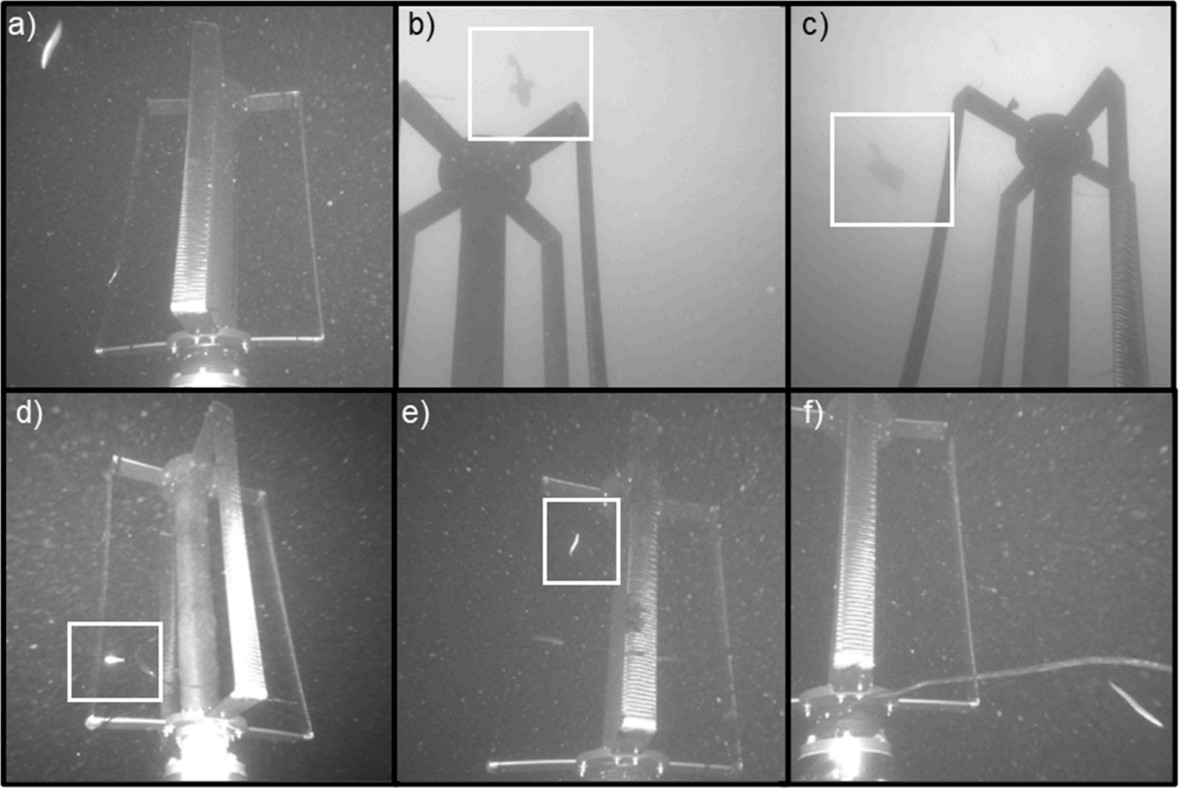

**Fig 6. Examples of individual fish when the turbine was rotating.** (a) Small fish drifting, head down, between AMP and rotor. (b) Large, unidentified fish swimming above rotor. (c) Flatfish drifting above moving rotor. (d) Perch-like species drifting with current between AMP and rotor. (e) Fish swimming within moving rotor. (f) Fish swimming away from rotor after collision.

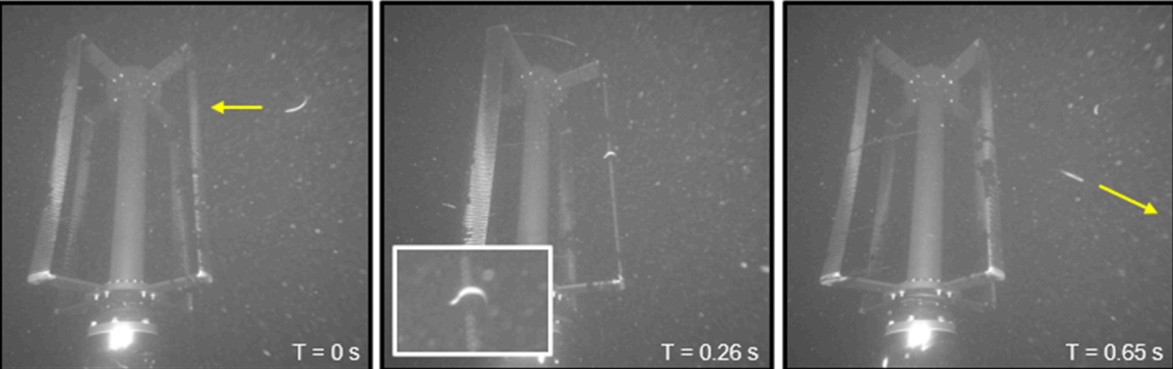

**Fig 7**. **Example of a collision between a fish and the turbine blades**. The yellow arrows approximate the direction the fish was moving in a given frame, and times are presented relative to the first frame. The inset in the middle frame shows a larger view of the collision seen on the blade on the right-hand side of the image.

The flow speed during this event was 1.5 m/s, and the fish was swimming with the currents before the collision. We note that a fifth instance of fish collision was observed with the stationary turbine (inflow speed of 0.9 m/s). The fish was drifting passively, without moving, when it collided with the blade and then swam away. Review of the concurrent acoustic camera data revealed that prior to two of the collision events (one while the turbine was moving during 1.0 m/s currents, and one while the turbine was stationary), a seal had pursued the fish towards the turbine. In both events, the seal abandoned the pursuit and changed directions before entering the optical camera field of view.

Annotated imagery showing four of the 50 cases of observed evasion behavior by fish, and one of a seal, are shown in Fig 8. Like collision events, fish were observed moving both with and against the currents when they encountered the turbine before evasion. Four types of evasion behavior were observed: 1) swimming then changing trajectory to move around the turbine rotor, 2) diving towards the seafloor or swimming upwards to pass under or over the rotor, 3) swimming against the current to move away from the rotor, or 4) evading the turbine blades and entering and then exiting the turbine rotor without collision. Evasion of the moving turbine was observed at flow speeds ranging from 1-2.2 m/s. In two events, seals were observed in the same image sequence, and pursued the fish towards the turbine before the fish exhibited evasion behavior. Review of the acoustic camera data identified that two of these fish evasion events were related to seal/fish predator/prey interactions where the seal was beyond the optical camera field of view. Additionally, in one event, a small perch-like fish exhibited evasion behavior while the turbine was stationary (0.5 m/s flow speed). The fish was observed passively drifting towards the Turbine Lander, then suddenly swam down and under the rotor.

## Fish schools

Nineteen total fish schools were recorded (Fig 9). Fish schools were observed both while the turbine was rotating and while it was stationary (Table 2). While the turbine was stationary, fish schools were observed dispersing and changing direction around the turbine rotor (e.g., Fig 9a, 9b). The five schools observed during turbine rotation were composed of unidentified small fish. In all five cases, the fish that encountered the turbine rotor effectively evaded it by swimming upstream and away from the rotor, ultimately passing between cameras and the turbine, or by diving down below the rotor (e.g., Fig 9d). Review of concurrent acoustic camera data indicated that the school shown in Fig 9d was considerably larger than was visible in the optical cameras and that most fish in the school avoided the rotor outside of the optical camera field of view. The individual fish associated with fish schools that evaded the rotor could not be readily counted and are therefore not included in the evasion/collision metrics presented for individual fish.

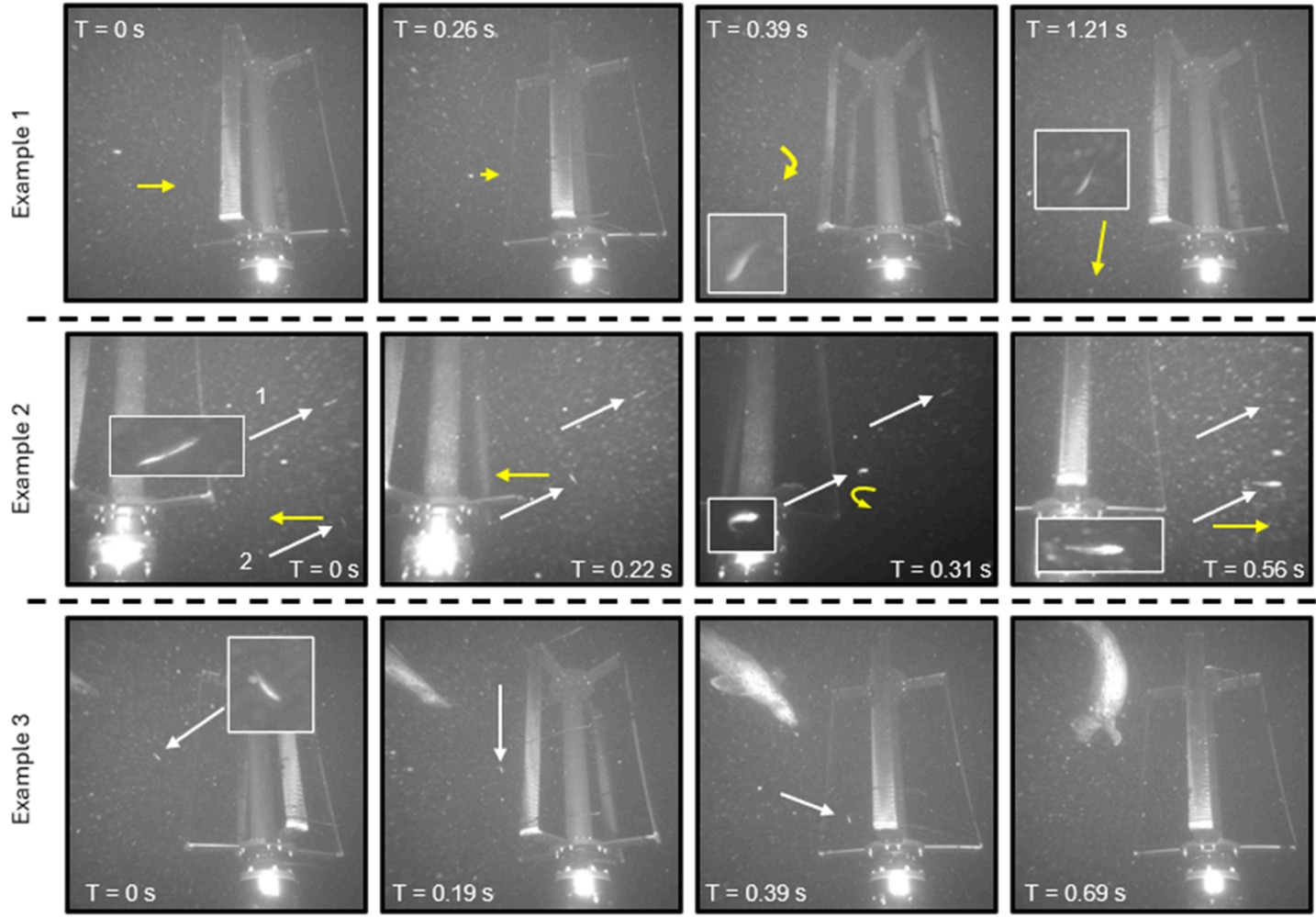

**Fig 8. Examples of evasion events while the turbine was operational.** In the images, the white arrows point to the fish and the yellow arrows approximate the direction of travel in optical images. Times are presented relative to the first frame. (Example 1) An evasion event with a single small fish. The fish approaches the rotor before turning and diving down away from the rotor. (Example 2) Two fish evading the turbine rotor using different tactics. Fish 1 swims against the current and moves out of the field of view. Fish 2, swimming with the current, executes a turn and swims away from the rotor. (Example 3) A seal pursuing a fish. The fish swims into and out of the rotor, evading a collision with the moving rotor in the process. As the seal approaches the moving rotor it stops pursuing the fish, slows down, and swims away from the rotor.

### Seabirds

A total of 406 unique events containing seabirds, including pigeon guillemots, double-crested cormorants, and possibly pelagic cormorants, were recorded (see examples in Fig 10). Of the observed seabirds, 74% (299) were identified as cormorants and 26% (105) were identified as pigeon guillemots. One event could not be attributed to either species because the bird was too close to the camera to identify distinguishing features.

Diving seabirds were exclusively observed when the turbine was not operating (Table 2), and were most frequently observed at high tide, when there were the longest periods of near-slack water (Fig 4). Seabirds were only detected at flow speeds greater than the minimum turbine cut-in speed of 0.9 m/s on two occasions. On 7 February, a pigeon guillemot was observed swimming towards the water surface between the turbine and the AMP when the flow speed was

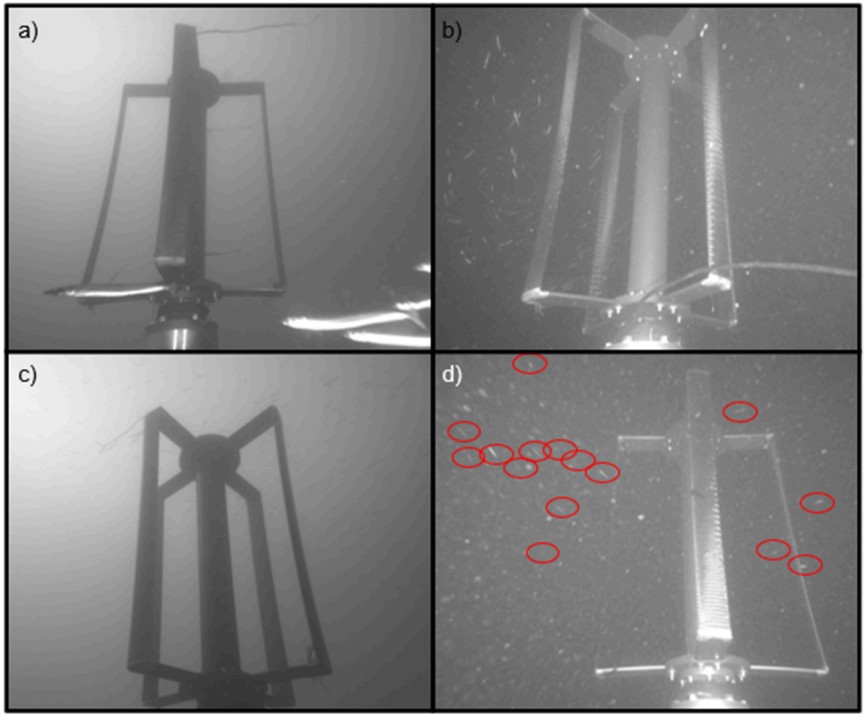

**Fig 9. Examples of fish schools.** (a–b) A small school of forage fish near the AMP. (b) A fish school near the rotor. Most fish swim up or around the backside of the rotor although several swim between the rotor and the camera. (c) A fish school near the surface while the turbine was stationary. (d) A fish school evading the turbine rotor at night while the turbine was rotating. Red circles highlight the individual fish.

approximately 0.9 m/s. The turbine cut-in speed was adjusted to 1 m/s at this time, so the turbine was not rotating. At the end of deployment, when the turbine was no longer operational (12 February), a cormorant was observed diving at a shallow angle (across the frame) in the background when the flow speed was 1.02 m/s and the turbine would have recently started to rotate had it not been shut down. All observations of diving seabirds were during the day, and seabirds were observed diving through the camera field of view towards the base of the Turbine Lander and resurfacing with prey (fish), indicating that they were likely foraging around the base of the turbine.

## Seals

Harbor seals were observed in 92 unique events (Table 2). Most seals (90%) were detected during periods when the turbine was not operating, and no instances of collision were observed. Seals were observed during all hours of the day, but more frequently when it was dark (58%; Fig 4). This is particularly notable given the lack of automatic detection models and limited sampling at night for much of the deployment. We also note that observations of seals were not evenly distributed in time; seals were observed on 44 unique days, with up to 22 events on a single night (3-4 November 2023).

Seals were observed encountering the turbine while it was moving on nine occasions. Three of these events were instances when the turbine was operating due to a system error and flow speeds were below 0.9 m/s, four were at flow speeds between 0.9 and 1 m/s, and two were at flow speeds between 1.3 and 1.4 m/s. Three types of seal behavior were observed while the turbine was rotating. In three events, the seal was observed in the background, and did not approach the moving turbine (Fig 11a). In four events, the seal approached the moving turbine rotor and swam in its wake with its head oriented towards the turbine before swimming away (Fig 11b). These interactions ranged in duration from a few

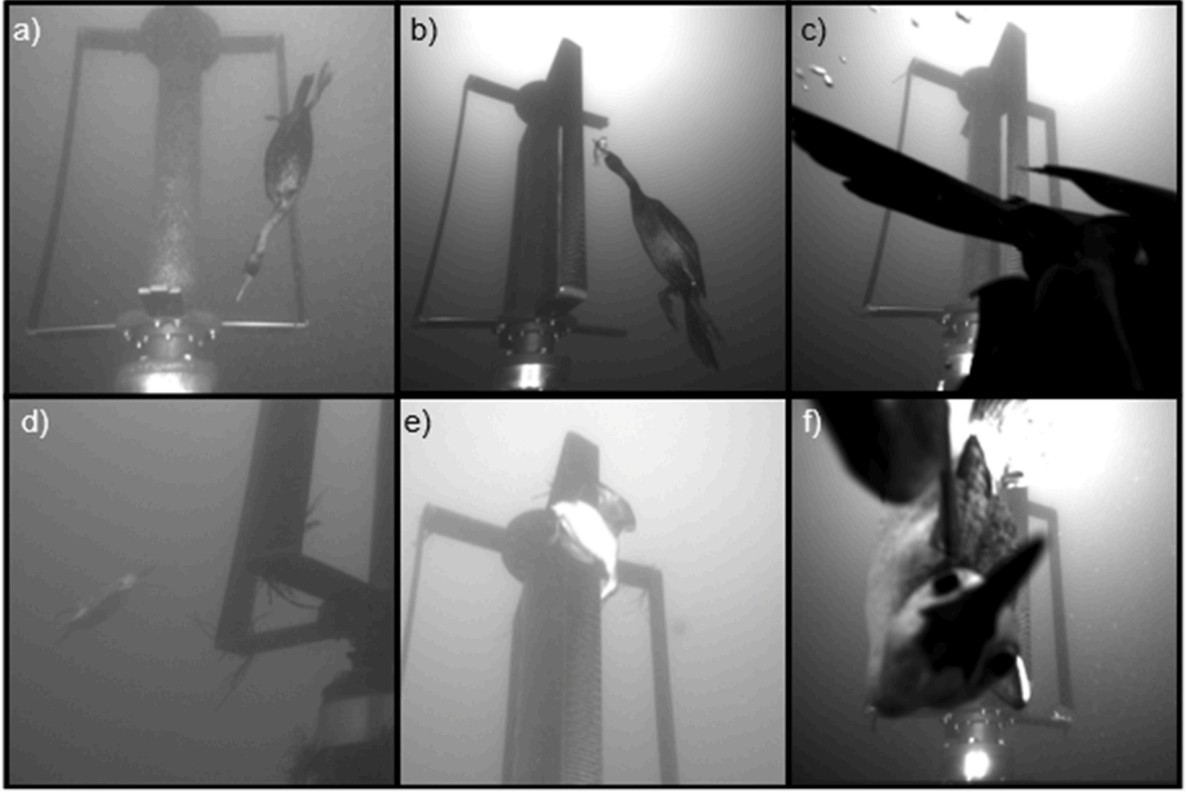

**Fig 10. Examples of seabirds**. (a–c) Cormorants diving, swimming to the surface, and presumably foraging in the vicinity of the AMP. (d-f) Pigeon guillemots swimming, picking at the rotor, and interacting with the AMP.

seconds to several minutes. Lastly, in two events, the seal was pursuing fish prey. In one of these events, upon encountering the turbine, the seal stopped pursuing the fish and changed trajectory to avoid colliding with the moving turbine rotor (i.e., evasion). The fish then passed through the rotor, evading the blades without collision (Fig 8). In the other case, the full event was not captured but the recorded frames show the fish near the turbine and swimming away from the seal while the seal swims away behind the rotor. Review of the acoustic camera data concurrent with fish evasion and collision events identified seven additional events where seals were in the vicinity of the turbine while it was rotating, but did not approach the turbine and therefore were not observed in the optical cameras (including the five fish events previously discussed).

While the turbine was stationary, seals were observed both swimming past the turbine without changing trajectory (i.e., no attraction) and directly interacting with the turbine structure or the AMP (i.e., attraction). Instances of seals interacting with the stationary turbine structure included swimming between the turbine rotor blades and shaft or simply approaching different parts of the rotor (Fig 11c, 11d).

### Comparison between continuously-acquired data and detected events

A comparison between the events in the overlapping period of data subsets 2 (daytime continuous data collection) and 3 (detected events) offers some insight into the performance of the real-time detection model for bird and seal detection. Due to a software bug introduced when continuous acquisition was enabled on 25 January, real-time detection models were not running reliably between 25 and 31 January. After the bug was resolved on 31 January, 94 of the 123 (76%)

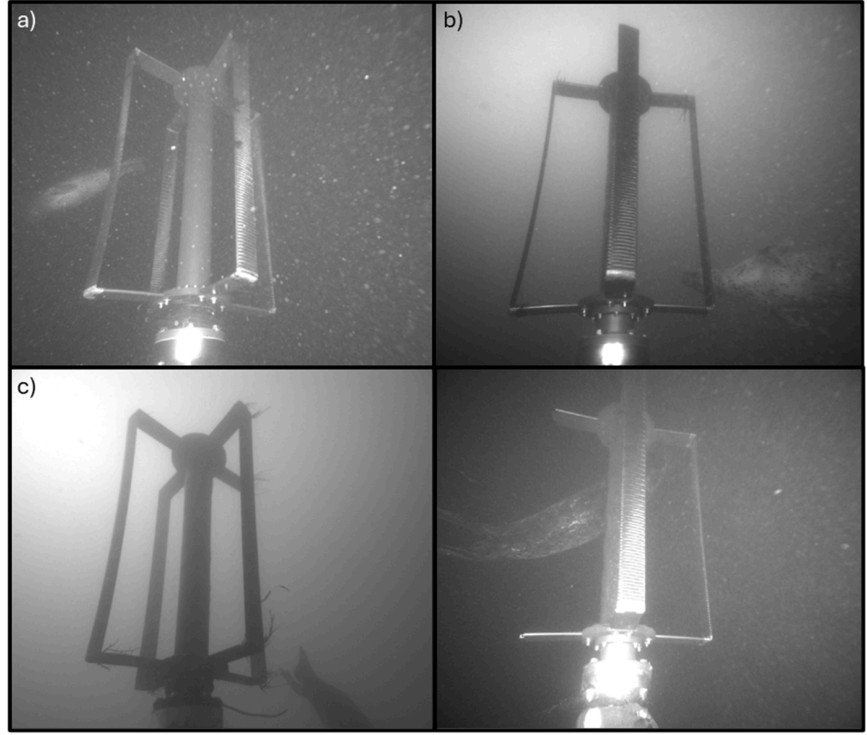

**Fig 11. Examples of seals.** a) Seal swimming behind the moving rotor at night. b) Seal approaching the moving rotor in the wake during the day. c) A seal diving towards the seabed. d) A seal entering the rotor's swept area and bending around the shaft. In both (c) and (d) the rotor was stationary.

events containing seabirds and 9 of the 11 (82%) events containing seals that were identified in the continuously acquired data comprising data subset 2 were also contained in data subset 3 (i.e., were detected by the real-time detection model). In 14 of the 29 bird events and both of the seal events that were missed by the detection model, the animal was only faintly visible in the background of the image beyond the turbine or near the edge of the camera field of view. This indicates that the detected events in data subset 3 likely contain most events where seabirds and seals approached the turbine rotor during periods with sufficient illumination for optical detection, though we cannot quantitatively assess performance throughout the deployment since the model was iteratively retrained. However, retraining was generally found to have a more significant effect on reducing the false positive rate (number of images archived that did not contain an animal of interest) than the false negative rate (number of animals missed).

In addition, Fig 4 shows that the trends in bird and seal presence with respect to time of day, flow velocity, and tidal elevation remain fairly consistent between the data subsets comprised of scheduled data acquisition (subsets 1 and 2) and the data subset comprised of detected events (subset 3). This indicates that, while we cannot quantitatively assess the total number of animals that encountered the turbine, our analysis of the conditions under which seabirds and seals encountered the turbine is likely representative.

## Discussion

### Limitations of this study and implications for collision risk

In the review of optical camera imagery spanning 109 days, we observed four instances of fish collision with the moving turbine (out of 229 individual fish and 5 fish schools observed encountering the moving turbine). No instances of collision with seabirds or marine mammals were observed. Several important limitations of this study must be considered when

interpreting the results. First, the real-time detection model that triggered data collection at a sufficiently high frame rate to resolve fine-scale fish behavior did not effectively detect fish. Therefore, our observations of fish offer insights into fish behavior around the turbine, but we cannot draw conclusions about trends in fish behavior or the total number of fish that encountered, evaded, or collided with the turbine. Second, sampling was limited at night, when the turbine was most frequently rotating due to local tidal forcing, indicating that many animal interactions with the moving turbine at night were likely not recorded. Finally, because optical camera observations are limited to the near-field, our results offer insight into the conditions under which animals encountered the moving turbine and their behavior around the turbine structure. We are not able to draw conclusions about avoidance behavior at ranges beyond the field of view of the cameras.

To our knowledge, this paper presents the first observations of fish collisions with a tidal turbine in the literature, though fish collision with a riverine turbine has previously been observed [31]. In three of the four cases of collision with the moving turbine, the fish appeared to swim away after the blade strike event. While we cannot quantify the total number of collisions that occurred, our observations indicate that the probability of collision for fishes is low (225 fish of the 229 fish observed encountering the moving turbine did not experience collision). Further, our observations indicate that even relatively small fish are capable of evasion at flow speeds exceeding 2 m/s. For these small fish, flow speeds during turbine operation typically exceed a value of 10 body lengths per second. This threshold corresponds to the upper limit of swimming speeds that most fish can achieve for short periods of time (burst swimming), while sustained swimming speed thresholds would be expected to be lower [32,33]. We hypothesize that the inability of small fish to swim against the current in a sustained manner may explain why most fish observed during turbine operation were small (i.e., large fish with stronger swimming capabilities may have avoided the turbine outside of the camera field of view). If true, burst swimming speeds, in conjunction with the speed of a turbine's blades, could be used to evaluate fish evasion capabilities. Finally, observations of seals pursuing fish immediately before those fish either collided with or evaded the turbine indicates that predator-prey interactions may be drivers of fine-scale fish behavior around a moving turbine.

The total number of fish encounters with the turbine that were identified by reviewers is likely a significant underestimate of the actual number of fish encounters during the reviewed time periods. Distinguishing between drifting plant matter and fish in the optical data was difficult for human reviewers because many fish were observed to be drifting without actively swimming. In these cases, lacking fish-like motion to unambiguously distinguish between plant matter and fish, we chose to limit positive identification of fish events to cases where the imagery was unambiguous due to fish-like characteristics (e.g., fins). In several cases, objects were observed that were not initially identified as fish when they entered the field of view, but were only determined to be fish after they exhibited evasion behavior when they approached the turbine rotor. We also speculate that fish were more readily discernible at night due to reflection of the direct light from the strobes. However, even at night, the orientation of the animals influenced detectability. Fish oriented perpendicular to the camera lens scattered relatively little light and appeared only as small dots with no discernible features unless their orientation changed during sampling.

Because evasion and collision both involve the fish actively moving, changing orientation, or interacting with a turbine blade, we feel confident that all of the clear instances of these interactions were logged. Instances of evasion that involved subtle changes to trajectory to avoid interactions may have been under-counted due to difficulties identifying the behavior. However, we believe it is reasonable to speculate that an order of magnitude more small fish were likely present in recorded images than were counted. If true, this would considerably decrease the fraction of fish encounters with the moving turbine that involved evasion or collision. Unfortunately, we cannot recommend any specific sampling or processing approaches to mitigate these biases in optical camera imagery unless conditions are suitable for color imagery. The challenges in optical detection could be mitigated, in part, through the use of acoustic cameras, though detection of fish in the vicinity of a moving turbine is not straightforward and there are many other challenges associated with detection of fish in acoustic camera imagery [34].

Because over 99% of seabirds were observed during periods with currents below the minimum turbine cut-in speed of 0.9 m/s, and all seabirds were detected when the turbine was not operating, our observations suggest that the risk of

collision for seabirds during this turbine deployment was low. This is consistent with studies of the foraging patterns of black guillemots in tidal channels conducted in Scotland, UK [20,21]. However, a study conducted in the same geographic region as the Turbine Lander deployment (Vancouver Island, Canada) found higher abundance, but no increase in diving behavior, of pigeon guillemots and pelagic cormorants during periods of high flow [35], and other species of seabirds have been shown to forage in strong currents [20]. We also only observed diving seabirds during daylight hours. This is consistent with previous observations of cormorants [36] and pigeon guillemots [37], but cormorants have been observed foraging at night in regions with seasonally low ambient light [38] and nocturnal foraging behavior has been reported for other species of guillemots [37]. The variability of trends in seabird behavior reported in the literature indicates that collision risk will vary between species and may vary between locations for the same species or family of seabirds. We note that subsurface observations are not necessary to determine temporal patterns in bird foraging; telemetry or visual surveys may provide much of the necessary information to assess this risk if the bird foraging depths are known.

Harbor seals most frequently encountered the turbine while it was stationary, though nine instances of seal interaction with the moving turbine were recorded. Three of these observations occurred during periods when the turbine was rotating at flow speeds below the turbine cut-in speed due to a software error. The fact that we detected most seals when the turbine was not operational is consistent with a recent study that used a multibeam sonar to monitor seal presence around a grid-scale tidal turbine in the Pentland Firth, Scotland [16]. When we observed seal encounters with the turbine while it was rotating, their behavior indicated that they were capable of evasion, even when pursuing prey. This would suggest that even though harbor seals may have exhibited attraction to the turbine, they were at low risk of collision. The highest risk may occur at turbine start-up if a seal is present and interacting with the stationary rotor. To mitigate this risk, devices could be programmed with a "soft start" wherein the turbine ramps up to its operating state slowly, giving any animals present the time to move away from the device. Alternatively, a deterrence device could be activated for a short period before turbine start-up.

Lastly, in considering the implications of this study for a broader understanding of collision risk, it is important to consider that design of a tidal turbine (e.g., cross-flow versus axial-flow) influences the risk of collision. Specifically, cross-flow turbines like the one in this study operate most efficiently at lower tip-speed ratios than axial-flow turbines [39]. For the majority of the deployment, the Turbine Lander blades moved at approximately two times the flow speed (tip-speed ratio of 2), resulting in peak blade speeds of less than 5 m/s. Axial flow turbines typically operate at tip-speed ratios between 4 and 6, meaning that, at the Sequim Bay test site, the tips of the blades of an axial flow turbine would be moving over 10 m/s during peak flow speeds, more than twice as fast as the cross-flow turbine studied here. Animals may be less capable of evading faster-moving blades, and the consequence of collision (i.e., severity of injury or mortality) with a faster moving blade would potentially be higher.

## Lessons learned and recommendations for future studies

While the optical camera imagery collected using the AMP offers many insights into how animals behaved in the vicinity of the Turbine Lander, we cannot draw conclusions about animal encounter rates or the probability of collision. There are several reasons for this, including our decision to focus our analysis on the optical camera imagery and the variable sampling strategies used throughout the deployment. In the sections that follow, we discuss lessons learned from this study and how they can inform future efforts.

**Trade-offs of optical vs. acoustic cameras.** As has been well established in previous work [40,41], no single sensor can provide all of the necessary information to characterize animal behavior around turbines. While optical cameras, like those used in this study, are the only sensor that can provide high-resolution information sufficient to characterize fine-scale animal behavior and facilitate species-level identification, they have limited range and are reliant on optical clarity and availability of light. While a single optical camera captured the entire Turbine Lander in its field of view, this is typically not possible for grid-scale turbines (e.g., [9]). Conversely, acoustic cameras can detect animals at the longer ranges

(tens of meters) necessary to monitor larger turbines and observe avoidance behavior. Acoustic cameras, depending on their operating frequency, can also operate in more turbid waters and without illumination, but their low resolution and frame rates (depending on range requirements) limit behavior classification capabilities for small animals, and even for larger animals at greater ranges. Our goal before deploying the Turbine Lander was to leverage the advantages of both sensor types for multi-sensor, 24-hour monitoring, but, we ultimately prioritized human review and development of automatic detection models for the optical camera data. While time and budget constraints were a factor, this approach was also driven by the fact that the position of the AMP on the Turbine Lander resulted in more compelling optical imagery. The optical cameras captured the entire turbine rotor, while the close range of the acoustic cameras meant that only a "slice" of the rotor was captured in the field of view. As a result of the narrow beamwidth at the range of the turbine, larger animals (i.e., seabirds and seals) took up a sufficiently large fraction of the beam to cause processing artifacts that partially obscured the image if the animal was near the turbine. Further, clutter in the acoustic camera imagery from the turbine structure and entrained sediment and plant matter made it difficult to interpret fine-scale animal behavior around the turbine.

Our focus on the optical cameras severely limited our observations at night. Because continuous operation of artificial illumination for nighttime monitoring with optical cameras, whether white or red light, can bias animal behavior [42, 43], we only enabled artificial illumination at night on a sparse duty cycle or when animals were detected in the acoustic camera data. A duty cycle is likely to miss interactions of interest, making acoustic camera-triggered artificial illumination ideal. However, deployment of automatic detection models on the acoustic camera imagery was not attempted until late in the deployment (February), and they did not perform well due to the large number of false positive detections resulting from the motion of the turbine and debris in the water column. Ultimately, better positioning of the acoustic cameras and the development of more effective strategies for real-time detection of animals in the presence of a moving turbine will be a high priority for future deployments of the AMP. While there are existing methods for the real-time, automatic detection of animals in acoustic camera data (e.g., [44]), the task is further complicated in the presence of a moving turbine structure because it introduces significant frame-to-frame variability even when animals of interest are not present. The development of effective algorithms for real-time acoustic camera detection of animals in the presence of a turbine will likely require a combination of advanced background subtraction methods, target tracking, and other classification models.

**Sampling strategies and automatic detection.** Many lessons were learned about how to balance conflicting priorities for data collection and about how to train and evaluate machine learning models for real-time animal detection throughout the Turbine Lander deployment. We initially adopted a high frame rate duty cycle to assess system performance and collect training data for real-time detection models (Fig 3). Our primary reason for this was that high frame rate data are required to evaluate fine-scale animal behavior, including collision and evasion, in either optical or acoustic camera imagery. For example, if an animal is moving with the currents at a flow speed of 2 m/s, and data are acquired at a low frame rate of 1 Hz, the animal will move 2 m between frames, a distance longer than the turbine rotor diameter. However, if data were acquired at 20 Hz, the animal would move 10 cm between frames, a distance similar to the chord length of one of the turbine blades. Because acquiring data continuously at such a high frame rate would accrue unwieldy volumes of data (over 175 GB per day if data were acquired at 20 Hz from one of the AMP cameras), we initially identified a duty cycle as the best solution for data collection before automatic detection was enabled.

In practice, we quickly identified that the duty cycle did not capture a sufficient number of events to train an automatic detection model or to gain a comprehensive understanding of the types of animals that were interacting with the turbine. Further, while the duty cycle produced many images of the animals that were recorded (e.g., 20 images of the same seal), it was preferable to have fewer images of many animals (e.g., two images each of 10 different seals) to build a more robust machine learning model for animal detection. Therefore, we switched to low frame rate, continuous acquisition on 8 November. This approach was effective in building a more robust training data set, and after just over a week, an automatic detection model was redeployed on 17 November. At this point, we disabled continuous data acquisition and began

duty cycle acquisition again. However, when faced with evaluating the false negative rate of the model (i.e., number of missed targets), we realized that the continuous, low frame rate data acquisition would again be preferable for the same reasons, and began continuous recording again on 25 January. Ultimately, our ability to quantitatively assess the performance of the models deployed in real-time before 25 January was limited, meaning that we cannot use our results to assess the rates of animal encounter, evasion, or collision.

Based on these experiences, and with the advantage of hindsight, we recommend the following strategy for future optical camera data collection campaigns with cabled instrumentation systems. Initially, low frame rate, continuous acquisition is recommended during daylight hours. If artificial illumination is available, a high duty cycle at night should be considered. While we acknowledge this may bias animal behavior, the benefits of sampling during periods when data are otherwise unavailable may outweigh the benefits of collecting an unbiased sample with few relevant observations. Further, sampling more frequently or even continuously at a high frame rate during periods when the turbine is rotating should also be considered, as these data are significantly more valuable than data collected while the turbine is stationary. Data should be reviewed and annotations used to train a detection model for identification of animal classes (e.g., fish, seabirds). This is time-intensive, but necessary, to enable subsequent reductions in data volume and associated processing requirements. If the model achieves suitable performance in testing, continuous data acquisition can be disabled, although low duty cycle data acquisition is still recommended. If the model is re-trained or updated during the deployment, the initial period of continuously acquired data, in addition to newly acquired duty-cycled data, can be used as a baseline to assess model performance and false negative rates. Depending on the species of interest and the image quality, in some cases, suitable model performance may not be achievable. For example, reliable detection of small fish was challenging for human reviewers in our dataset, and it is unlikely that a machine learning algorithm will be able to detect them reliably. In this case, high duty cycles or continuous data acquisition are recommended. A similar strategy would be recommended for acoustic cameras, but continuous data acquisition could be utilized during all hours of the day.

When evaluating the performance of detection models, we recommend using different evaluation metrics than are used for many applications of machine learning. In this application, the machine learning model is being used to identify periods of data to archive for human review, not to count the total number of frames that animals are identified in. For example, if a seal enters the field of view, there may be frames where only part of the seal is visible that might be missed by automatic detection algorithms. However, if the seal is detected by the model later in the sequence, data stored in a ring buffer can be archived to capture the entire event. This indicates that frame-by-frame assessment of model performance is not the most representative metric for model performance. Rather, model performance should be evaluated on an event basis - an event can be considered "captured" if the animal was detected in at least one frame that would be archived and flagged for human review. Further, for many applications of machine learning, correct and incorrect detections are weighted equally, but for this application, models should be tuned to favor a high true positive rate over a low false positive rate. The "cost" of an incorrect detection is additional human review and data storage, while the "cost" of a missed event is potentially lost knowledge of animal interactions of interest. As long as the cost imposed by false positives is acceptable, which would typically be measured by the time to archive and review them, relatively high false positive rates are not inherently problematic.

## Conclusions

While there are several limitations to the presented data, the observations of animal interactions with an operational tidal turbine reported in this study provide new insight into animal behavior around tidal turbines. Seabirds were likely at low risk for collision, given that they were frequently observed when the turbine was stationary but were never observed while it was operating. While this result may be limited to the specific species and site, it is the most comprehensive set of observations of seabird interactions with a tidal turbine to date. Seals were occasionally observed while the turbine was rotating, but exhibited strong swimming capabilities indicating that they are capable of evading collision. While four fish

collision events with the moving turbine were observed, more than 10 times more fish were observed evading the moving turbine blades, including during periods when flow speeds exceeded 2 m/s. Further, many more fish were likely present in the imagery that encountered the turbine without collision but were difficult or impossible to identify.

## Supporting information

**S1 Species Table. List of all animal families and species observed.** SpeciesTable.xlsx includes all observed animals, reviewer confidence in taxonomic classification, and qualitative notes on general behavior observed for each species or family. Confidence was assessed subjectively based on the clearly identifiable characteristics including size, shape, and patterns. The notes describe the confidence of the classification and frequency with which the animals were observed.
(XLSX)

**S2 Event Data. Metadata for all annotated events.** Events.zip contains CSV files listing the start time, end time, animal class, flow speed, and ambient light conditions for each annotated event. Qualitative notes on animal species and behavior are also included.
(ZIP)

**S3 Videos. Corresponding videos for figures.** Corresponding_videos.csv contains the file name of the video file corresponding to each figure in the paper containing camera imagery. All videos can be found at [30].
(CSV)

**S4 Detailed Description of Human Review Process. Written overview of image review process** Review Process.pdf provides a more detailed overview of the human review process for optical images described in the manuscript.
(PDF)

## Acknowledgments

The authors wish to thank Lenaïg Hemery for assistance with species identification, Garrett Staines for reviewing an early version of this manuscript, the Pacific Northwest National Laboratory dive team for maintenance of the Turbine Lander, and MarineSitu team members James Joslin, Emily Paine, and Sadie Kass for their contributions to AMP data collection strategy, system deployment and recovery, and data management and data delivery systems. Lastly, we would like to thank the reviewers for their thoughtful reviews of the manuscript.

## Author contributions

**Conceptualization:** Christopher Bassett.

**Data curation:** Alexa Runyan, Jood M. Almokharrak, Lucy G. Kao, Lillian M. Ovall, Suni A. McMillen.

**Formal analysis:** Emma Cotter, Christopher Bassett, Mitchell Scott.

**Funding acquisition:** Christopher Bassett.

**Investigation:** Emma Cotter.

**Methodology:** Christopher Bassett.

**Project administration:** Christopher Bassett, Alexa Runyan.

**Software:** Paul Murphy, Mitchell Scott.

**Validation:** Paul Murphy.

**Visualization:** Emma Cotter.

**Writing – original draft:** Emma Cotter, Christopher Bassett.

**Writing – review & editing:** Paul Murphy, Mitchell Scott, Alexa Runyan, Jood M. Almokharrak, Lucy G. Kao, Lillian M. Ovall, Suni A. McMillen.

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
