## [Decision Letter · Decision Letter 0]

1 Oct 2025

PONE-D-25-24729Observations of marine animal interactions with a small tidal turbinePLOS ONE

Dear Dr. Cotter,

Thank you for submitting your manuscript to PLOS ONE. After careful consideration, we feel that it has merit but does not fully meet PLOS ONE’s publication criteria as it currently stands. Therefore, we invite you to submit a revised version of the manuscript that addresses the points raised during the review process.

We look forward to receiving your revised manuscript.

Kind regards,

Jianhong Zhou

Staff Editor

PLOS ONE

Journal Requirements:

[The development and deployment of the Turbine Lander was sponsored by the Naval Facilities Engineering and Expeditionary Warfare Center (NAVFAC) under Naval Sea Systems Command contract N00024-20-F8708. Environmental monitoring data management was funded by the U.S. Department of Energy Water Power Technologies Office (WPTO) TEAMER program, and analysis of the collected data was funded by WPTO (award EE0007827). Development of the machine learning and data management tools used by MarineSitu for this work was supported by WPTO through the Small Business Innovation and Research project "Modular Instrumentation and Automated Data Processing for Marine Energy Monitoring" (Award DE-SC0021845).]. 

[The development and deployment of the Turbine Lander was sponsored by the Naval Facilities Engineering and Expeditionary Warfare Center (NAVFAC) under Naval Sea Systems Command contract N00024-20-F8708. Environmental monitoring data management was funded by the U.S. Department of Energy Water Power Technologies Office (WPTO) TEAMER program, and analysis of the collected data was funded by WPTO (award EE0007827). The authors wish to thank Lenaïg Hemery for assistance with species identification, Garrett Staines for reviewing an early version of this manuscript, the Pacific Northwest National Laboratory dive team for maintenance of the Turbine Lander, and MarineSitu team members James Joslin, Emily Paine, and Sadie Kass for their contributions to AMP data collection strategy, system deployment and recovery, and data management and data delivery systems.]

[The development and deployment of the Turbine Lander was sponsored by the Naval Facilities Engineering and Expeditionary Warfare Center (NAVFAC) under Naval Sea Systems Command contract N00024-20-F8708. Environmental monitoring data management was funded by the U.S. Department of Energy Water Power Technologies Office (WPTO) TEAMER program, and analysis of the collected data was funded by WPTO (award EE0007827). Development of the machine learning and data management tools used by MarineSitu for this work was supported by WPTO through the Small Business Innovation and Research project "Modular Instrumentation and Automated Data Processing for Marine Energy Monitoring" (Award DE-SC0021845).]

[The authors have declared that no competing interests exist.].   

We note that one or more of the authors are employed by a commercial company: MarineSitu

5. Please amend the manuscript submission data (via Edit Submission) to include author Emma Cotter.

6. Please amend your authorship list in your manuscript file to include author Emma DeWitt Cotter.

7. We note that Figures 1, 6, 7, 8, 9, 10, and 11 in your submission may contain copyrighted images. All PLOS content is published under the Creative Commons Attribution License (CC BY 4.0), which means that the manuscript, images, and Supporting Information files will be freely available online, and any third party is permitted to access, download, copy, distribute, and use these materials in any way, even commercially, with proper attribution. For more information, see our copyright guidelines: http://journals.plos.org/plosone/s/licenses-and-copyright.

1. You may seek permission from the original copyright holder of Figures 1, 6, 7, 8, 9, 10, and 11 to publish the content specifically under the CC BY 4.0 license.

In the figure caption of the copyrighted figure, please include the following text: “Reprinted from [ref] under a CC BY license, with permission from [name of publisher], original copyright [original copyright year].

8. We note that Figure 2 in your submission contains map images which may be copyrighted. All PLOS content is published under the Creative Commons Attribution License (CC BY 4.0), which means that the manuscript, images, and Supporting Information files will be freely available online, and any third party is permitted to access, download, copy, distribute, and use these materials in any way, even commercially, with proper attribution. For these reasons, we cannot publish previously copyrighted maps or satellite images created using proprietary data, such as Google software (Google Maps, Street View, and Earth). For more information, see our copyright guidelines: http://journals.plos.org/plosone/s/licenses-and-copyright.

1. You may seek permission from the original copyright holder of Figure 2 to publish the content specifically under the CC BY 4.0 license. 

Reviewers' comments:

Reviewer's Responses to Questions

**Comments to the Author**

1. Is the manuscript technically sound, and do the data support the conclusions?

Reviewer #1: Yes

Reviewer #2: Yes

2. Has the statistical analysis been performed appropriately and rigorously?

Reviewer #1: Yes

Reviewer #2: N/A

3. Have the authors made all data underlying the findings in their manuscript fully available?

Reviewer #1: No

Reviewer #2: No

4. Is the manuscript presented in an intelligible fashion and written in standard English?

Reviewer #1: Yes

Reviewer #2: Yes

5. Review Comments to the Author

Reviewer #1: No -for date. There is a repository (see detailed notes below) listed in the paper, but data don't seem to have been uploaded yet. Once this is resolved, I'm happy that the answer to Q3 above will be yes

General Comments

A welcome paper both providing useful insights into technologies for monitoring animal movement around tidal energy systems and also highlighting some of the difficulties in detecting small animals close to large bits of moving machinery. Some of the conclusions are a not very conclusive, but that is the nature of these data and I think it’s important that it’s published, in part as a statement of how difficult this type of work is, and to inform people that there is not (anytime soon at least) going to be a magical system that can process these types of data automatically (i.e. cheaply) as the tidal energy industry expands.

I’ve made quite a few comments on the text below. There are a few contradictions that need clarifying and a few points that I’m happy for the authors to ignore.

Specific Comments

Introduction

Good and useful review on knowledge to date in lines 1 – 72.

L 57: “harbor seal presence” This study could not tell the difference between harbor and grey seals, both present in the area, so it’s just “seal presence”, or “harbor and grey seal presence”.

L76. What do the terms meso and macro mean ?. So far as I can tell, meso scale avoidance is the same as the definition of avoidance that you took earlier from Garavelli, and micro avoidance is what you’ve earlier called evasion. I think you should stick to one set of terms. Avoidance and Evasion were quite nicely defined earlier, so maybe best to stick with them, so this might read: “Further, most studies to date have studied animal avoidance at larger scales, but there have been few reports of fine scale evasion behavior in the immediate vicinity of a tidal turbine”

Methods

L94: ‘in several’ or ‘on several’ ?

L112 – 115: Should this be in a separate section on Ethics and Permits ?

Around L137: Can you say more about how the fields of view of the sonars were organised. From figure 1, it looks as though the two BlueView’s have the same vertical orientation, but are about 90 deg separate horizontally, giving a combined swath of over 180 degrees ? Why didn’t you orientate them the same horizontally and at different vertical angles to cover the full height of the device ? Also, did you sample these at the same time as the Gemini ? Was there interference – although they are at different frequencies, I’m guessing quite bad side lobes ?

L160 or earlier at around 134: can be be more explicit about when you operated the lights ? There is some information at L165, and again at line 175, but was the strategy ? Were they only used after model detections ?

L168 ‘… using their …’ is this referring to MarineSitu or Ultralytics ? Do you mean ‘deployment’ software, or ‘detection’ software ?

At L171 you say the model reliably detected seals. At L180 and again at line 185 you say that it did not perform well for that it performed poorly. There is a contradiction here that needs to be resolved.

L182: At this point, you start to bring results into the methods section. While I can see that you do need some generalities about the results here, to give context to the iterative methods, try to keep specific results as much as possible to the Results section.

L306: The video repository Basse & Cotter, https://mhkdr.openei.org/submissions/599 appears to still be in progress and doesn’t yet have any videos available.

Discussion

Fish collision rate. It seems that you rate of collision is still about 2%, which is not entirely insignificant. I know that for marine mammals, regulators are currently using a range of evasion / avoidance values, though 98% seems to be one that’s popular. Are there any similar numbers for fish, that are currently being used in collision risk models, that you could compare your numbers to ? I’m guessing that you’re staying away from doing this because you think the 4 collisions is accurate, but the 229 fish is a vast underestimate.

L466. You state here that the real time detection model was not trained to detect fish. I thought that one of the main things you’d been doing was retraining and redeploying a model on the early data to specifically detect fish. This sentence seems to be in direct contradiction of the methods.

Line 484: Can you clarify this please. You making a comparison between a speed and a length, which is not really possible.

Lines 491 to 494. Can you clarify whether the difficulty is with AI models, human observers, or both please. Do you have numbers on how many ambiguous targets there were as well as the 229 positive identifications. I think that quantifying the uncertainty in all these measurements is important in guiding how future studies might be implemented and what level of data they might provide to regulators and other interested parties. I think you need to expand the results section a bit

L548 soft start – or use of a deterrence device for a few seconds prior to startup ?

L558: I’m pretty sure the tip speed of the turbines in the Pentland Firth is around 13m/s (9m radius, 14rpm), in a 3-4m/s current.

L562. Do you know the mass of the blades in your system ? For the big turbines, it’s so much bigger than the mass of an animal that the animal is going to get a hard whack and barely slow the blades at all. For your turbine it looks as though the blades would probably (or have?) come of worse in a collision because they are probably lighter than a seal. Probably too much of a rabbit hole to discuss this, but you might put the mass of the system into the introduction somewhere so that people can do their own math on this if they are that way inclined.

L583: I think you could remove “particularly for fish”, or say “for small or animals at any range, and even for larger animals at greater ranges”.

L628: 175GByte per day. Hard yes, but can be done, particularly at this pilot stage of developing these systems.

660: “Once the model achieves …”. What if they don’t ? Everyone seems to be taking that attitude that the AI will eventually be good enough. Even in your own training data, you say you’ve a lot of targets that you don’t know what they are – fish or debris. How do you even handle them in model training ? While I agree that with time, you’ll make better algorithms, and you’ll be able to train them with more data, I don’t actually believe they will be that much better than the current ones that are inadequate in many ways. Perhaps you need to stress in these final paragraphs the importance of keeping a human in the loop, and of having review software that can allow the human to efficiently examine these events. Getting that software right might be more beneficial that continually bang

Reviewer #2: Cotter et al., reviewed marine animal interactions with a marine tidal turbine in Sequim Bay in the Pacific Northwest over a period of 109 days using optical imagery. Due to the testing nature of the deployment the data capture methods varied throughout the study but three subsets of data were annotated for interactions with fish, schools of fish, seals, and seabirds. There were ~1000 interactions recorded total and the most common behavior observed was evasion with only 4 collisions, all with fish. There was an interesting observation that half of these collisions likely resulted from being chased by a seal. Even though the ability to do any statistical analysis was limited, the findings are significant for a growing field and with the potential to inform future deployments. The authors provide valuable reflection on the limitations of their dataset and present learnings that will be appreciated in the field. The paper is well-written and the technical aspects were well-described for a general science audience.

Some more comments below and in the attached pdf.

Authors have put N/A under ethics statement which also includes field work. Although the work does not involve collection of samples, surely there would have been permissions/permitting process to install the tidal turbine and associated instrumentation for the work.

Associated data appear to be available and well-documented.

Title could have more impact - the results indicate there were minimal animal collisions with the turbine which could be highlighted right at the front. Currently, have to read ¾ down the abstract to find out this info. I can understand that the authors might consider this overselling their results but they present a good dataset and understanding of the limitations to make more of a statement.

Authors have done a good job describing how the data collection changed throughout the deployment – Figure 3 was very helpful in this regard.

The methods left me wondering how the 5 annotator events were collated into a final event log? What was the variability/range between annotators? Presentation of some of the methods/data concerning this and discussion around challenges, things to consider could be a valuable addition.

The presentation of Table 2, Fig 4, and Fig 5 could be more impactful. Currently, describes the table/figures and is a missed opportunity to present some of the trends around the environmental variables, e.g., most bird/seal events are at low flow speeds and high tidal range.

6. PLOS authors have the option to publish the peer review history of their article (what does this mean?). If published, this will include your full peer review and any attached files.

Reviewer #1: **Yes: **Douglas Gillespie

Reviewer #2: **Yes: **Catherine Kim

---

## [Author Response · Author response to Decision Letter 1]

17 Oct 2025

Reviewer comments are addressed in the attached response to comments document.

---

## [Decision Letter · Decision Letter 1]

23 Nov 2025

Observations of marine animal interactions with a small tidal turbine

PONE-D-25-24729R1

Dear Dr. Cotter,

We’re pleased to inform you that your manuscript has been judged scientifically suitable for publication and will be formally accepted for publication once it meets all outstanding technical requirements.

Kind regards,

James K. Sheppard

Academic Editor

PLOS ONE

Additional Editor Comments (optional):

This revised manuscript incorporates clear, targeted revisions that directly address the minor concerns of both reviewers and the editor, substantially improving clarity, transparency, and framing. The authors strengthened the Methods by more fully describing the sensor configuration, lighting, and annotation workflow, and clarified how ambiguous detections and model limitations (especially for fish) were handled. They also corrected some confusing terminology and phrasing (e.g., around “avoidance/evasion” and swimming speed vs. body length), and polished the presentation of the Results section so that the narrative is consistent with the documented capabilities and limitations of the instruments and models. In addition, the authors resolved data-availability, ethics and image permission issues. The Discussion and Abstract now more carefully distinguish observed encounter fractions from broader collision-risk probabilities and explicitly acknowledge undercounting and field-of-view constraints, which aligns the conclusions with the strength of the evidence.

I believe these changes adequately address the reviewers’ concerns. Consequently, I refer my recommendation for publication in PLOS One to the Editor-in-Charge for the final decision on this manuscript.

Reviewers' comments:

Reviewer's Responses to Questions

**Comments to the Author**

1. If the authors have adequately addressed your comments raised in a previous round of review and you feel that this manuscript is now acceptable for publication, you may indicate that here to bypass the “Comments to the Author” section, enter your conflict of interest statement in the “Confidential to Editor” section, and submit your "Accept" recommendation.

Reviewer #1: All comments have been addressed

2. Is the manuscript technically sound, and do the data support the conclusions?

Reviewer #1: Yes

3. Has the statistical analysis been performed appropriately and rigorously?

Reviewer #1: N/A

4. Have the authors made all data underlying the findings in their manuscript fully available?

Reviewer #1: Yes

5. Is the manuscript presented in an intelligible fashion and written in standard English?

Reviewer #1: Yes

6. Review Comments to the Author

Reviewer #1: One remaining typo !

Line 242 included in data (missing ‘in’)

All comments on previous draft have been fully addressed.

7. PLOS authors have the option to publish the peer review history of their article (what does this mean?). If published, this will include your full peer review and any attached files.

Reviewer #1: **Yes: **Douglas Gillespie

---

## [Editor Report · Acceptance letter]

PONE-D-25-24729R1

PLOS ONE

Dear Dr. Cotter ,

I'm pleased to inform you that your manuscript has been deemed suitable for publication in PLOS ONE. Congratulations! Your manuscript is now being handed over to our production team.

Kind regards,

on behalf of

Dr. James K. Sheppard

Academic Editor

PLOS ONE